# Goal-Oriented Skill Abstraction for Offline Multi-Task Reinforcement Learning

**Jinmin He** [1 2]  **Kai Li** [1 2]  **Yifan Zang** [1 2]  **Haobo Fu** [3]  **Qiang Fu** [3]  **Junliang Xing** [4]  **Jian Cheng** [1 5]

## Abstract

Offline multi-task reinforcement learning aims to learn a unified policy capable of solving multiple tasks using only pre-collected task-mixed datasets, without requiring any online interaction with the environment. However, it faces significant challenges in effectively sharing knowledge across tasks. Inspired by the efficient knowledge abstraction observed in human learning, we propose Goal-Oriented Skill Abstraction (GO-Skill), a novel approach designed to extract and utilize reusable skills to enhance knowledge transfer and task performance. Our approach uncovers reusable skills through a goal-oriented skill extraction process and leverages vector quantization to construct a discrete skill library. To mitigate class imbalances between broadly applicable and task-specific skills, we introduce a skill enhancement phase to refine the extracted skills. Furthermore, we integrate these skills using hierarchical policy learning, enabling the construction of a high-level policy that dynamically orchestrates discrete skills to accomplish specific tasks. Extensive experiments on diverse robotic manipulation tasks within the MetaWorld benchmark demonstrate the effectiveness and versatility of GO-Skill.

## 1. Introduction

Deep reinforcement learning (RL) has made significant progress in recent decades, demonstrating its effectiveness in diverse domains such as game playing (Mnih et al., 2015; Ye et al., 2020) and robotic control (Lillicrap et al., 2016; Levine et al., 2016). However, its deployment in real-world scenarios is often limited by the high costs and risks associated with directly interacting with the environment. To

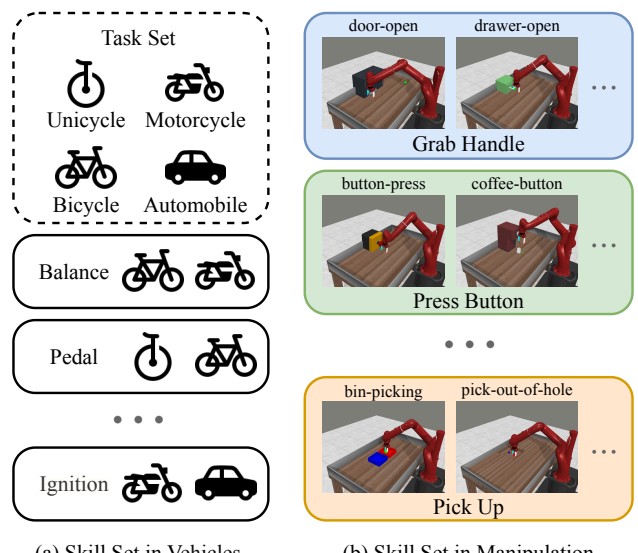

*Figure 1.* Skill sets shared across different domains. (a) Skill set for human learning to operate four types of vehicles: bicycles and motorcycles share the skill of balancing on two wheels; unicycles and bicycles share the pedal-driven skill; motorcycles and automobiles share the ignition-to-move skill. (b) Skill set in robotic manipulation tasks: grabbing various handles, pressing buttons at different heights, and picking up objects from diverse locations.

address these challenges, offline RL (Levine et al., 2020) has emerged as a data-driven approach, enabling policy learning entirely from pre-collected datasets, thereby eliminating the need for real-time interaction. Recent advancements in offline RL have expanded its applicability to single-task domains, such as robotics (Kalashnikov et al., 2018; 2021) and healthcare (Guez et al., 2008; Killian et al., 2020). In contrast, offline multi-task reinforcement learning (MTRL) seeks to effectively master a set of offline RL tasks using the pre-collected task-mixed dataset. Joint multi-task learning generally enhances adaptability and performance compared to training each task individually.

A critical challenge in offline MTRL lies in effectively sharing knowledge between tasks. Recent research has proposed some approaches by leveraging network parameter sharing, incorporating carefully designed architectures (Yang et al., 2020; He et al., 2024b; Hu et al., 2024), task-specific

---

[1]C²DL, Institute of Automation, Chinese Academy of Sciences [2]School of Artificial Intelligence, University of Chinese Academy of Sciences [3]Tencent AI Lab [4]Tsinghua University [5]AiRiA. Correspondence to: Kai Li <kai.li@ia.ac.cn>, Junliang Xing <jlxing@tsinghua.edu.cn>.

*Proceedings of the 42ⁿᵈ International Conference on Machine Learning*, Vancouver, Canada. PMLR 267, 2025. Copyright 2025 by the author(s).

representations (Sodhani et al., 2021; Lee et al., 2022; He et al., 2023), and tailored optimization procedures (Chen et al., 2018; Yu et al., 2020a; Liu et al., 2021). However, these methods predominantly operate at a lower level, focusing on action-based imitation learning, which contrasts with how humans typically approach learning. Humans tend to abstract knowledge more effectively by identifying and summarizing common strategies or patterns from past experiences into a set of reusable skills. Instead of learning solely at the action level, they dynamically combine these high-level skills to address specific tasks. For instance, when learning to operate four different types of vehicles—unicycle, bicycle, motorcycle, and automobile—humans distill shared skills as illustrated in Figure 1(a). This human-inspired mechanism of skill abstraction offers a fresh perspective for offline MTRL, suggesting that agents could similarly distill and leverage skills to enhance learning efficiency, as demonstrated in robotic manipulation scenarios in Figure 1(b).

Motivated by this, we propose Goal-Oriented Skill Abstraction (GO-Skill), a novel method for discovering a skill library from offline task-mixed datasets and learning a skill-based policy capable of addressing multiple tasks through skill combination. First, we introduce a skill model that leverages goal-oriented representations to extract reusable skills, then uses vector quantization to construct a discrete skill library. These skills act as low-level policies, enabling interaction with the environment. To address the potential class imbalance dilemma, where widely applicable skills have abundant data while task-specific skill data is limited, we incorporate a skill enhancement phase. Finally, we utilize the hierarchical policy learning approach, in which a skill-based policy is learned using the discrete skill space as the action space. This enables the effective combination of these skills to solve specific tasks.

GO-Skill not only mirrors the way humans learn but also has the potential to improve sample efficiency in offline MTRL significantly. Specifically, offline trajectory data is not exclusively composed of expert demonstrations but may include sub-optimal or random trajectories. Although sub-optimal trajectories may provide limited direct benefits for a specific task, they often contain transferable skill sub-trajectories that can be valuable for other tasks. GO-Skill excels at extracting these reusable skill fragments from less-than-optimal data, enabling their application across diverse tasks. Furthermore, skill abstraction shortens the decision-making horizon for the skill-based policy to accomplish tasks, simplifying the policy learning process. This approach improves sample efficiency and contributes to a more robust and effective multi-task learning process.

We evaluate GO-Skill in the MetaWorld benchmark (Yu et al., 2020b), which consists of 50 robotic manipulation tasks. Our method demonstrates significant improvements over existing offline MTRL algorithms. Additionally, we conduct comprehensive ablation studies to assess each GO-Skill component's contribution.

## 2. Preliminaries

**Offline Multi-Task Reinforcement Learning.** We aim to learn $N$ tasks simultaneously, each represented as a Markov Decision Process (MDP) (Bellman, 1966; Puterman, 2014). Specifically, each task $i \in \mathbb{T}$ is defined by the tuple $\langle \mathcal{S}, \mathcal{A}, \mathcal{P}_i, \mathcal{R}_i, d_i \rangle$, where $\mathcal{S}$ denotes the state space, $\mathcal{A}$ the action space, $\mathcal{P}_i : \mathcal{S} \times \mathcal{A} \to \mathcal{S}$ the environment transition dynamics, $\mathcal{R}_i : \mathcal{S} \times \mathcal{A} \to \mathbb{R}$ the reward function, and $d_i$ the initial state distribution. In line with common MTRL setting (He et al., 2024b; Hu et al., 2024), all tasks share the same state and action spaces but differ in transition dynamics, reward functions, and initial state distributions. The goal of the MTRL agent is to maximize the average expected return across all tasks, expressed as $\mathbb{E}_{i \in \mathbb{T}} \mathbb{E} \left[ \sum_{t=0}^{\infty} \mathcal{R}_i(\boldsymbol{s}_t, \boldsymbol{a}_t) \right]$. In the offline setting (Levine et al., 2020), instead of collecting data through environment interactions, each task has a static dataset $\mathcal{D}_i = \{(\boldsymbol{s}, \boldsymbol{a}, \boldsymbol{s}', r)\}$, consisting of trajectories from the environment.

**Prompt Decision Transformer.** The Transformer architecture (Vaswani et al., 2017), widely used in natural language processing (Devlin et al., 2019) and computer vision (Carion et al., 2020), has demonstrated superior performance over traditional RNN-based models. Recently, it has been applied to solve RL problems, leveraging its efficiency and scalability when dealing with long sequential data. The Decision Transformer (DT) (Chen et al., 2021) for offline RL forms the policy learning as a sequential modeling problem. Prompt-DT (Xu et al., 2022) extends the DT approach by incorporating prompt-based techniques, enabling few-shot generalization to new tasks. It introduces trajectory prompts, which consist of return-to-go, state and action tuples $(\hat{r}^*, \boldsymbol{s}^*, \boldsymbol{a}^*)$ from task-specific trajectory demonstrations. During training with offline collected data, Prompt-DT utilizes $\tau_{i,t}^{\text{input}} = (\tau_i^*, \tau_{i,t})$ as input for each task $i$, where $\tau_i^*$ represent $K^*$-step trajectory prompt and $\tau_{i,t}$ denotes the most recent $K$-step history. This is formally represented as:

$$
\tau_{i,t}^{\text{input}} = \left( \hat{r}_{i,1}^*, \boldsymbol{s}_{i,1}^*, \boldsymbol{a}_{i,1}^*, \ldots, \hat{r}_{i,K^*}^*, \boldsymbol{s}_{i,K^*}^*, \boldsymbol{a}_{i,K^*}^*, \right. \\
\left. \hat{r}_{i,t-K+1}, \boldsymbol{s}_{i,t-K+1}, \boldsymbol{a}_{i,t-K+1}, \ldots, \hat{r}_{i,t}, \boldsymbol{s}_{i,t}, \boldsymbol{a}_{i,t} \right). \tag{1}
$$

The model's prediction head, connected to a state token $\boldsymbol{s}_j$, is designed to predict the corresponding action $\boldsymbol{a}_j$. For continuous action spaces, the training objective is to minimize the mean-squared loss, defined as:

$$
\mathcal{L}_{DT} = \mathbb{E}_{\tau_{i,t}^{\text{input}} \sim \mathcal{D}_i} \left[ \sum_{j=t-K+1}^{t} \| \boldsymbol{a}_{i,j} - \pi(\tau_i^*, \tau_{i,j}) \|_2^2 \right]. \tag{2}
$$

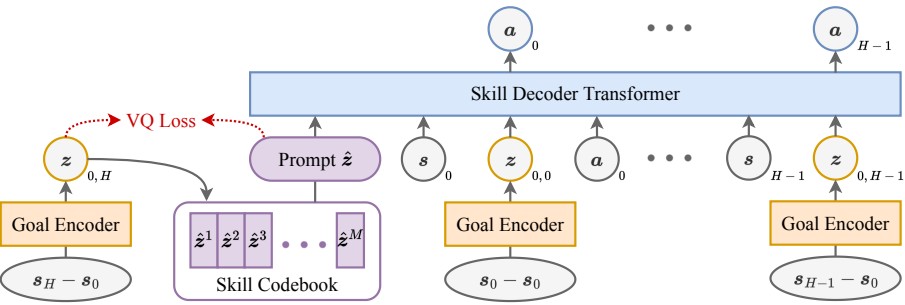

*Figure 2.* Illustration of the skill model in GO-Skill, which comprises three key components: (1) *Goal Encoder* maps the trajectory-level difference between states to a latent embedding space; (2) *Skill Codebook* distills the $H$-horizon goal embedding into a discrete skill set via vector quantization; (3) *Skill Decoder* reconstructs actions with skill prompt and history sequence using a Transformer architecture.

**Vector Quantization.** The vector quantization (VQ) module (Van Den Oord et al., 2017) has been used for many applications by learning discrete latent representations. This VQ module $\mathcal{F}$ maintains a latent embedding space (codebook) $\boldsymbol{M} \in \mathbb{R}^{C \times L}$, composed of $C$ vectors $\boldsymbol{e}_i \in \mathbb{R}^L$. Given an encoder output $\boldsymbol{z} \in \mathbb{R}^L$, it selects the discrete latent variable as $j = \arg\min_i \|\boldsymbol{z} - \boldsymbol{e}_i\|_2$. Based on the discrete code $j$, we get the query vector $\mathcal{F}(\boldsymbol{z}) = \boldsymbol{e}_j$ from the codebook. During training, the codebook and encoder output are optimized with the following loss,

$$\mathcal{L}_{VQ}(\boldsymbol{z}, \boldsymbol{e}_j) = \|\text{sg}[\boldsymbol{z}] - \boldsymbol{e}_j\|_2^2 + \alpha \|\boldsymbol{z} - \text{sg}[\boldsymbol{e}_j]\|_2^2, \quad (3)$$

where $\text{sg}[\cdot]$ is stop-gradient operator, and $\alpha$ adjusts the balance of minimizing the discrepancies between $\boldsymbol{z}$ and $\mathcal{F}(\boldsymbol{z})$.

## 3. Method

Motivated by knowledge abstraction learning illustrated in Figure 1, we propose Goal-Oriented Skill Abstraction (GO-Skill). This approach addresses two key challenges: 1) Extracting reusable skills from the offline task-mixed dataset. 2) Leveraging these skills effectively to tackle diverse tasks.

### 3.1. Goal-Oriented Skill Extraction

GO-Skill extracts a discrete library of reusable skills from the offline dataset by learning a skill model, which leverages trajectories across diverse tasks to build a robust foundation. We define a skill as a sub-sequence of trajectories with a fixed horizon $H$. Distinguishing from common sequence modeling approaches (Chen et al., 2021; Xu et al., 2022; Lee et al., 2022) that predict action based on past (return-to-go, state, action) experience, our skill model excludes reward signals (reward and return-to-go) from the skill definition, as the reward function varies across tasks, even for identical trajectories. While a skill trajectory may not yield high rewards in the context of a particular task, it can be invaluable for mastering essential skills required in other tasks. As shown in Figure 2, the skill model in GO-Skill

consists of three parts: goal encoder, skill quantization, and skill decoder.

**Goal Encoder.** The skill achieves a dynamic transfer from the initial state $\boldsymbol{s}_t$ to the target state $\boldsymbol{s}_{t+H}$. To capture this process in a task-agnostic manner, we introduce the goal as a representation of this dynamic transfer. For the $H$-horizon sub-trajectory, we define the goal encoder $\mathcal{G} : \mathcal{S} \to \mathcal{Z}$, which maps the trajectory-level dynamic transfer between the initial state $\boldsymbol{s}_t$ and the target state $\boldsymbol{s}_{t+H}$ to a latent embedding space $\mathcal{Z}$. Specifically, the goal embedding is expressed as,

$$\boldsymbol{z}_{t,H} = \mathcal{G}(\boldsymbol{s}_{t+H} - \boldsymbol{s}_t). \quad (4)$$

Here, we represent the $H$-horizon dynamic transfer directly in terms of state variation. For more complex state situation, a goal-oriented representation can be defined. For example, in navigation tasks, the goal may primarily depend on the agent's position. In cases where the state is represented as an image, we can leverage the pre-trained vision-language models (Nair et al., 2022; Ma et al., 2023) to convert the state into a relevant embedding representation. Notably, we do not use action sequences to represent dynamic transfer, as different action sequences can result in the same transfer outcome. This distinction is analyzed in our experiments Section 5.3.

**Skill Quantization.** To distill the large number of subsequences into a discrete library of skills, we use vector quantization (Van Den Oord et al., 2017) to learn discrete latent skill representations with goal embeddings $\boldsymbol{z}_{t,H}$. Specifically, the acquired collection of skill embeddings (codebook) is denoted as $\mathbb{Z} = \{\hat{\boldsymbol{z}}^{(1)}, \hat{\boldsymbol{z}}^{(2)}, \ldots, \hat{\boldsymbol{z}}^{(M)}\}$, where $M$ is the predefined number of skills. The skill embedding is generated through the VQ module $\mathcal{F} : \mathcal{Z} \to \mathbb{Z}$ with the $H$-horizon goal embedding from the Goal Encoder:

$$\hat{\boldsymbol{z}}_t = \mathcal{F}(\boldsymbol{z}_{t,H}) = \arg\min_j \left\| \boldsymbol{z}_{t,H} - \hat{\boldsymbol{z}}^{(j)} \right\|_2, \quad (5)$$

and the skill codebook $\mathbb{Z}$ is optimized using the VQ loss as described in Equation 3.

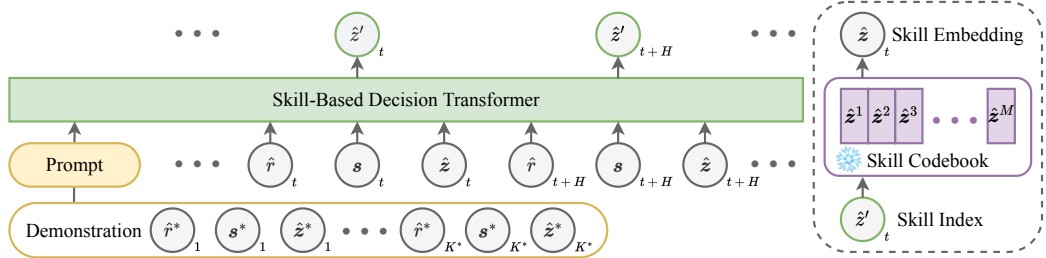

*Figure 3.* Illustration of the skill-based policy in GO-Skill. The policy, employing the Transformer architecture, uses the discrete skill space as the action space. Its inputs include (return-to-go, state, skill) prompts, and $H$-step interval historical trajectory. At each decision point, the policy predicts a skill index, which is subsequently mapped to its corresponding skill embedding via the skill codebook.

**Skill Decoder.** The skill model serves as low-level policy for interacting with the environment. We run the history sub-trajectory along with the skill embedding through the skill decoder to obtain an approximate reconstructed action. Beyond utilizing historical states and actions, we extend the goal to include historical sequences. The goal encoder is adapted to within a horizon of $H$, extending Equation 4 as,

$$z_{t,h} = \mathcal{G}(s_{t+h} - s_t), \quad 0 \le h \le H. \tag{6}$$

We refer to the goal embeddings $z_{t,h}(h < H)$, representing the difference between the current state $s_{t+h}$ and the initial state $s_t$ in the history sequence, as the *reached-goal*, which facilitates the agent in assessing the progress toward completing the current skill.

In summary, the skill decoder is formalized as $\mathcal{P} : \mathbb{Z} \times \Phi_{\mathcal{S},\mathcal{Z},\mathcal{A}} \to \Delta\mathcal{A}$, where $\Phi_{\mathcal{S},\mathcal{Z},\mathcal{A}}$ denotes the (state, reached-goal, action) trajectory space, and $\Delta\mathcal{A}$ stands for the probability distributions space over $\mathcal{A}$. In our implementation, we use the Transformer architecture (Vaswani et al., 2017),

$$\tilde{a}_{t'} \sim \mathcal{P}(\hat{z}_t, s_{\le t'}, z_{\le t'}, a_{<t'}), \quad t \le t' < t+H, \tag{7}$$

where $s_{\le t'} = s_t, s_{t+1}, \dots, s_{t'}$ is the state history, $z_{\le t'} = z_{t,0}, z_{t,1}, \dots, z_{t,t'-t}$ is the reached-goal history, $a_{<t'} = a_t, a_{t+1}, \dots, a_{t'-1}$ is the action history, and the skill embedding constitutes the prompt for this decoder transformer.

**Skill Enhancement.** As the skill model undergoes training, the set of skills gradually stabilizes. However, a notable challenge, skill class imbalance, arises. The decoder tends to favor mastering broadly applicable skills with abundant data while neglecting task-specific skills often constrained by limited data availability. Since every skill is essential, including task-specific ones, we propose a skill enhancement phase after the skill set has been fixed. Once the skill set converges, we freeze the goal encoder and the skill codebook. Next, we partition the dataset by skill class $\mathcal{D}^{(i)} = \{\tau'^{(i)}\} = \{(s_{t:t+H}, z_{t,0:H}, a_{t:t+H-1})\}$. To mitigate the imbalance, we apply a resampling strategy that uniformly samples across skill classes, thereby enhancing the training of the skill decoder.

We summarize the training procedure for skill extraction and enhancement in Algorithm 1.

### 3.2. Goal-Oriented Skill Policy Learning

The skill model can be treated as a frozen low-level policy to interact with the environment at the action level. To accomplish tasks, we employ a hierarchical policy learning scheme. In this scheme, a high-level skill-based policy is learned within a discrete skill space to select the appropriate skills.

**Dataset Preprocessing.** We partition the trajectory data into segments of $H$ time steps and annotate each segment with a skill index $\hat{z}'$ and skill embedding $\hat{z}$, which are generated by the frozen goal encoder and skill codebook. The resulting skill-based policy trajectory data is represented as $\mathcal{T} = \{(\hat{r}_T, s_T, \hat{z}_T, \hat{z}'_T)\}$, where $T$ denotes decision points at intervals of $H$ time steps. The same preprocessing is applied to prompt demonstration trajectories.

**GO-Skill Policy Learning.** As shown in Figure 3, we train a skill-based decision transformer using Prompt-DT (Xu et al., 2022) as the policy to select a skill at intervals of $H$ time steps,

$$\pi(\cdot | \hat{r}^*, s^*, \hat{z}^*, \hat{r}_{\le T}, s_{\le T}, \hat{z}_{<T}), \tag{8}$$

where $\hat{r}^* = \hat{r}_1^*, \hat{r}_2^*, \dots, \hat{r}_{K^*}^*$ represents the return-to-go prompt, $s^* = s_1^*, s_2^*, \dots, s_{K^*}^*$ represents the state prompt, $\hat{z}^* = \hat{z}_1^*, \hat{z}_2^*, \dots, \hat{z}_{K^*}^*$ represents the skill prompt. Additionally, $\hat{r}_{\le T}$ is the return-to-go history, $s_{\le T}$ is the state history, $\hat{z}_{<T}$ is the skill history. The skill-based policy outputs a skill index $\hat{z}'_T$, which is then used to query the codebook for the corresponding skill embedding $\hat{z}_T$. To address the imbalance in skill classes, we incorporate focal loss (Lin et al., 2017) into training,

$$\mathcal{L}_{FL}(\pi) = -(1 - \pi(\hat{z}'_T|\cdot))^\gamma \log(\pi(\hat{z}'_T|\cdot)), \tag{9}$$

where $\gamma \ge 0$ is the tunable focusing parameter. The GO-Skill policy learning process is outlined in Algorithm 2. Since the skill decoder and the skill-based policy operate independently, they are trained in parallel in implementation.

**Algorithm 1** GO-Skill Extraction and Enhancement

> **Input:** skill horizon $H$, skill set size $M$, goal encoder $\mathcal{G}$, skill VQ module $\mathcal{F}$, skill decoder $\mathcal{P}$, learning rate $\eta$
> **Initialize:** the parameters of the network $\theta_{\mathcal{G},\mathcal{F},\mathcal{P}}$
> // Skill Extraction
> **for** each extraction training iteration **do**
>     **for** each task $i \in \mathbb{T}$ **do**
>         get minibatch $\tau_i = (\boldsymbol{s}_{i,t:t+H}, \boldsymbol{a}_{i,t:t+H-1})$
>     **end for**
>     get batch data $\tau = \{\tau_i\}_{i \in \mathbb{T}}$
>     get goal embeddings $\{\boldsymbol{z}_{t,t'-t} = \mathcal{G}(\boldsymbol{s}_{t'} - \boldsymbol{s}_t)\}_{t'=t}^{t+H}$
>     build batch goal-oriented trajectory training data $\tau' = (\boldsymbol{s}_{t:t+H-1}, \boldsymbol{z}_{t,0:H-1}, \boldsymbol{a}_{t:t+H-1})$
>     get skill embedding $\hat{\boldsymbol{z}}_t = \mathcal{F}(\boldsymbol{z}_{t,H})$
>     predict action $\tilde{\boldsymbol{a}}_{t'} \sim \mathcal{P}(\hat{\boldsymbol{z}}_t, \boldsymbol{s}_{\leq t'}, \boldsymbol{z}_{\leq t'}, \boldsymbol{a}_{<t'})$
>     $\mathcal{L}_{VQ} = \|\text{sg}[\boldsymbol{z}_{t,H}] - \hat{\boldsymbol{z}}_t\|_2^2 + \alpha \|\boldsymbol{z}_{t,H} - \text{sg}[\hat{\boldsymbol{z}}_t]\|_2^2$
>     $\mathcal{L}_{MSE} = \frac{1}{H} \sum_{t'=t}^{t+H-1} \|\tilde{\boldsymbol{a}}_{t'} - \boldsymbol{a}_{t'}\|_2^2$
>     $\theta_{\mathcal{G},\mathcal{F},\mathcal{P}} \leftarrow \theta_{\mathcal{G},\mathcal{F},\mathcal{P}} - \eta\nabla(\mathcal{L}_{MSE} + \mathcal{L}_{VQ})$
> **end for**
> // Skill Enhancement
> build skill class dataset $\mathcal{D}^{(i)} = \{\tau'^{(i)}\}$
> **for** each enhancement training iteration **do**
>     **for** each skill index $i = 1, \dots, M$ **do**
>         get minibatch $\tau'^{(i)} = \left(\boldsymbol{s}_{t:t+H}^{(i)}, \boldsymbol{z}_{t,0:H}^{(i)}, \boldsymbol{a}_{t:t+H-1}^{(i)}\right)$
>     **end for**
>     get batch data $\tau' = \{\tau'^{(i)}\}_{i=1}^M$
>     get skill embedding $\hat{\boldsymbol{z}}_t = \mathcal{F}(\boldsymbol{z}_{t,H})$
>     predict action $\tilde{\boldsymbol{a}}_{t'} \sim \mathcal{P}(\hat{\boldsymbol{z}}_t, \boldsymbol{s}_{\leq t'}, \boldsymbol{z}_{\leq t'}, \boldsymbol{a}_{<t'})$
>     $\mathcal{L}_{MSE} = \frac{1}{H} \sum_{t'=t}^{t+H-1} \|\tilde{\boldsymbol{a}}_{t'} - \boldsymbol{a}_{t'}\|_2^2$
>     $\theta_{\mathcal{P}} \leftarrow \theta_{\mathcal{P}} - \eta\nabla\mathcal{L}_{MSE}$
> **end for**

**Algorithm 2** GO-Skill Policy Learning

> **Input:** skill-based policy $\pi$, learning rate $\eta$
> **Initialize:** the parameters of the policy $\theta_\pi$
> **for** each training iteration **do**
>     **for** each task $i \in \mathbb{T}$ **do**
>         get minibatch $\mathcal{T}_i = (\hat{r}_{i,T:T+K}, \boldsymbol{s}_{i,T:T+K}, \hat{\boldsymbol{z}}_{i,T:T+K})$
>         get prompt $\mathcal{T}_i^* = \left(\hat{r}_{i,1:K^*}^*, \boldsymbol{s}_{i,1:K^*}^*, \hat{\boldsymbol{z}}_{i,1:K^*}^*\right)$
>     **end for**
>     get batch data $\mathcal{T} = \{\mathcal{T}_i^*, \mathcal{T}_i\}_{i \in \mathbb{T}}$
>     skill-based policy predict $\pi(\cdot|\mathcal{T})$
>     $\mathcal{L}_{FL} = -(1 - \pi(\hat{z}_{T'}'|\cdot))^\gamma \log(\pi(\hat{z}_{T'}'|\cdot))$
>     $\theta_\pi \leftarrow \theta_\pi - \eta\nabla\mathcal{L}_{FL}$
> **end for**

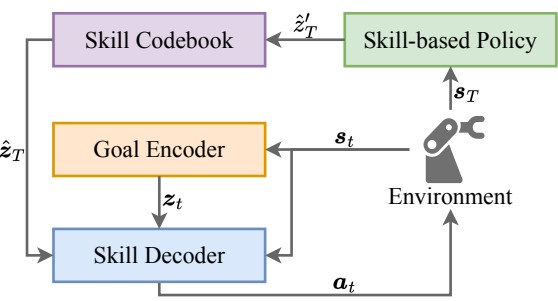

*Figure 4.* Deploying GO-Skill as a control policy to interact with the environment. $T$ represents the decision points of the skill-based policy at intervals of $H$ time steps, while $t$ indicates the time steps where the skill model predicts actions within the $H$-horizon.

icy are fine-tuned in parallel, enabling skill enhancement and policy adaptation to new tasks.

## 3.3. Deployment

Deploying GO-Skill as a policy is illustrated in Figure 4. At every $H$-step interval, the skill-based policy selects the skill index $\hat{z}_T'$ for the next $H$ time steps, based on the given prompt and historical trajectories. The predicted skill index $\hat{z}_T'$ is then mapped to its corresponding skill embedding $\hat{\boldsymbol{z}}_T$ by querying the skill codebook. During the $H$-step period, the initial state $\boldsymbol{s}_{t_0}$ (where $\boldsymbol{s}_{t_0} = \boldsymbol{s}_T$) and the current state $\boldsymbol{s}_t$ are processed by the goal encoder to generate the reached-goal $\boldsymbol{z}_{t_0,t-t_0}$. Finally, using the skill embedding $\hat{\boldsymbol{z}}_T$ and historical information within the current skill horizon $(\boldsymbol{s}_{t_0:t}^{(i)}, \boldsymbol{z}_{t_0,0:t-t_0}^{(i)}, \boldsymbol{a}_{t_0:t-1}^{(i)})$, the skill decoder generates actions $\boldsymbol{a}_t$ to interact with the environment.

## 3.4. Fine-Tuning for New Tasks

GO-Skill is also well-suited for transfer scenarios. First, we construct the skill-based policy trajectory dataset described in Section 3.2 using the frozen goal encoder and skill codebook. Next, the skill decoder and skill-based pol-

## 4. Related Work

**Offline Reinforcement Learning.** Offline RL focuses on learning policies from static offline datasets without requiring direct interaction with the environment. Unlike traditional RL, offline RL faces the challenges of distribution shift between the collected data and the learned policy (Fujimoto et al., 2019). To address this, prior research has explored constrained and regularized policy updates that limit deviations from the behavior policy (Kumar et al., 2020; Kostrikov et al., 2021; Ghosh et al., 2022). A notable direction in offline RL is conditional sequence modeling, in which actions are predicted from sequences of past experiences. This approach, which is grounded in supervised learning, inherently constrains the learned policy within the distribution of the behavior policy while optimizing for future trajectory metrics (Chen et al., 2021; Xu et al., 2022; Yamagata et al., 2023). Additionally, recent studies have incorporated diffusion models into offline RL (Janner et al., 2022; Chen et al., 2023; Wang et al., 2023), leveraging the strengths of generative modeling to represent policies or

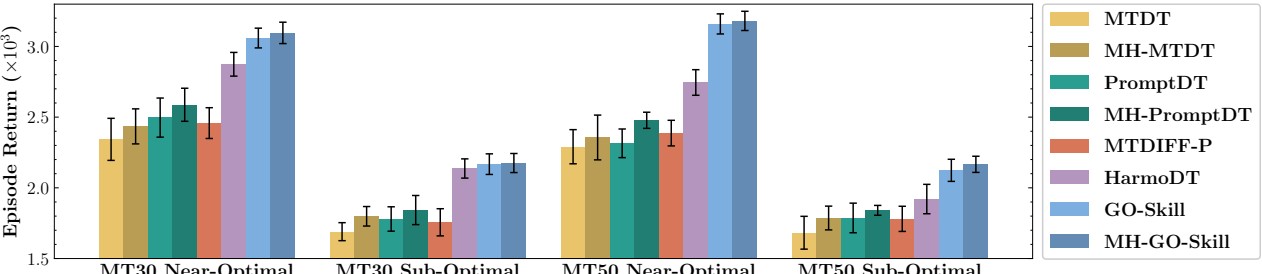

*Figure 5.* Average episode return across 5 random seeds on the MetaWorld benchmark with four different setups. The error bars represent the standard deviation across the 5 seeds. Each method is trained for 1e5 iterations, and each task is evaluated over 50 episodes.

dynamics, yielding competitive or superior performance.

**Multi-Task Reinforcement Learning.** Multi-task learning is a training paradigm that enhances generalization by leveraging domain knowledge shared among related tasks (Caruana, 1997). MTRL builds on this concept to discover shared knowledge across tasks by concurrently learning multiple reinforcement learning tasks. A straightforward approach to MTRL involves task-conditional modeling, using techniques like conditional representation (Sodhani et al., 2021), conditional sequence modeling (Xu et al., 2022; Lee et al., 2022), or conditional diffusion model (He et al., 2023). Due to gradient conflicts arising from simultaneously optimizing a shared model across diverse tasks, some methods (Chen et al., 2018; Yu et al., 2020a; Liu et al., 2021) reformulate MTRL as a multi-objective optimization problem, modifying the update process to mitigate these conflicts. Alternatively, other approaches avoid fully sharing the network across tasks, instead dynamically selecting sub-networks through mechanisms like distinct heads (D'Eramo et al., 2020), specialized modules (Yang et al., 2020; He et al., 2024b), or task-specific masks (Hu et al., 2024), enabling the construction of tailored task policies. While most approaches implicitly incorporate knowledge sharing within network parameters, GO-Skill leverages explicit knowledge abstraction to enhance generalization.

**Skill in Reinforcement Learning.** To utilize offline datasets for long-horizon tasks, recent research has explored hierarchical skill learning approaches within the realms of online reinforcement learning (Pertsch et al., 2021; 2022) and imitation learning (Hakhamaneshi et al., 2021; Nasiriany et al., 2023; Du et al., 2023). Pertsch et al. (2021) introduced a hierarchical skill framework designed to enhance the adaptation to downstream tasks by incorporating pre-learned skill priors that inform a high-level policy. At the same time, Hakhamaneshi et al. (2021) developed a semi-parametric method within a hierarchical framework, emphasizing its application to few-shot imitation learning. In online MTRL, Shu et al. (2018) uses a hierarchical policy that decides when to directly use a previously learned policy and when

to acquire a new one. Moreover, He et al. (2024a) employs a guide policy that leverages the control policies of other tasks to facilitate exploration and guide the target task learning. In our work, we also utilize skill embedding technique, but GO-Skill adapts to conditional sequence modeling with an innovative goal-oriented skill abstraction in offline MTRL.

## 5. Experiments

### 5.1. Environments and Baselines

**Environments.** Our experiments are evaluated on the MetaWorld benchmark (Yu et al., 2020b), an MTRL benchmark consisting of 50 robotic manipulation tasks using the sawyer arm in the MuJoCo environment (Todorov et al., 2012). We consider two setups: *MT30*, which includes a suite of 30 tasks, and *MT50*, which comprising all 50 tasks. In line with recent studies (Yang et al., 2020; He et al., 2024b), we extend all tasks to a random-goal setting. Following He et al. (2023), the dataset are sourced from SAC-Replay (Haarnoja et al., 2018) ranging from random to expert experiences, divided into two compositions: *Near-Optimal* containing the complete experiences and *Sub-Optimal* consisting of the initial 50% experiences. Details on environment specifications are available in the Appendix A.

**Baselines.** We compare GO-Skill against six baselines. (i) **MTDT**: Extension of DT (Chen et al., 2021) to multi-task settings. (ii) **MH-MTDT**: Extension of MTDT apart from independent heads for each task to predict action. (iii) **PromptDT** (Xu et al., 2022): Builds upon DT, leveraging trajectory prompts for generalization to multi-task learning. (iv) **MH-PromptDT**: Extension of PromptDT apart from independent heads for each task to predict action. (v) **MTDIFF-P** (He et al., 2023): A diffusion-based method combining Transformer architectures and prompt learning for generative planning in offline MTRL settings. (vi) **HarmoDT** (Hu et al., 2024): A DT-based approach that learns task-specific masks for each task, mitigating the adverse effects of conflicting gradients during parameter sharing.

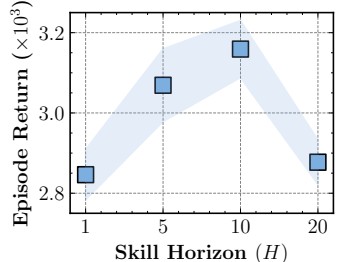 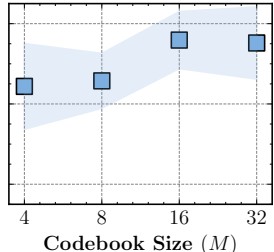 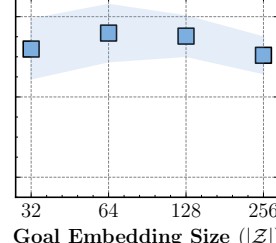 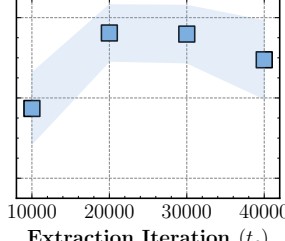

*Figure 6.* Ablation study on the hyper-parameter of GO-Skill under the Near-Optimal setting in the MT50 setup. Default parameter values are set as skill horizon $H = 10$, codebook size $M = 16$, code embedding size $|\mathcal{Z}| = 64$, and extraction iteration $t_e = 3e4$ (30% of total iteration). Each ablation varies a single parameter while keeping all others fixed at their default values to isolate its effect on performance.

*Table 1.* Ablation study on the skill model of GO-Skill under Near-Optimal and Sub-Optimal settings in the MT50 setup. *RG*: The skill decoder utilizes the reached-goal history. *VQ*: The skill set is discretized using the VQ module. *AE*: The skill embedding is encoded based on the action sub-sequence rather than the goal.

| RG | VQ | AE | Near-Optimal | Sub-Optimal |
|----|----|----|----|----|
| ✓ | ✓ | ✗ | $3159.1_{\pm71.0}$ | $2123.8_{\pm78.1}$ |
| ✗ | ✓ | ✗ | $2952.1_{\pm92.3}$ | $2041.5_{\pm83.5}$ |
| ✓ | ✗ | ✗ | $2905.0_{\pm92.7}$ | $1999.0_{\pm82.4}$ |
| ✗ | ✓ | ✓ | $2797.5_{\pm141.7}$ | $1947.9_{\pm126.2}$ |

*Table 2.* Ablation study on the skill imbalance challenge under Near-Optimal and Sub-Optimal settings in the MT50 setup.

| Focal Loss | Resampling | Near-Optimal | Sub-Optimal |
|----|----|----|----|
| ✓ | ✓ | $3159.1_{\pm71.0}$ | $2123.8_{\pm78.1}$ |
| ✗ | ✓ | $3048.4_{\pm90.9}$ | $2092.7_{\pm71.0}$ |
| ✓ | ✗ | $2962.3_{\pm89.4}$ | $2012.9_{\pm109.2}$ |

### 5.3. Ablation Study

**Ablation on Skill Model.** Table 1 delineates our ablation study on three key components of the skill model. First, incorporating the skill decoder with reached-goal history enables the agent to better assess its progress toward completing the current skill. Second, we explore defining the skill space as a continuous space, where the skill-based policy directly outputs the skill embedding, rather than selecting a skill index from the discrete space. The result shows that, similar to how humans summarize skills, using discrete sets of actions facilitates faster and more intuitive learning of skill utilization. Finally, to highlight the benefits of goal-oriented skills, we use continuous actions as input to the skill encoder and apply the skill quantization. The result demonstrates that goal-oriented skill significantly enhances the agent's ability to extract and master effective skills.

**Ablation on Skill Imbalance.** Some skills are broadly applicable across most tasks, while some are specific to a smaller subset of tasks, resulting in a skill class imbalance challenge. To address this, we employ a resampling strategy during the skill enhancement phase for the skill model. For the skill-based policy, we utilize focal loss during training. Table 2 presents the results of our ablation study on these two methods. The findings indicate that the skill model's learning is more significantly impacted by class imbalance, and the resampling strategy effectively enhances the agent's ability to master each skill.

**Ablation on Hyper-Parameters.** We conduct comprehensive ablation studies to establish an empirical strategy for

### 5.2. Main Results

Similar to MH-MTDT and MH-PromptDT, we also implement a multi-headed variant of GO-Skill. However, instead of the head division at the task level, **MH-GO-Skill** uses the multi-head architecture in the skill model, and the skill-based policy model still fully shares parameters.

As shown in Figure 5, GO-Skill demonstrates its effectiveness in offline MTRL settings, achieving superior performance compared to existing state-of-the-art approaches. It excels in handling both sub-optimal and near-optimal datasets. Specifically, it effectively extracts valuable information from sub-optimal trajectories and accurately emulates optimal behaviors when higher-quality data is available. Notably, the advantages of GO-Skill become increasingly pronounced as the number of tasks increases. It is also worth noting that baselines typically perform worse on MT50 than on MT30 when comparing the different setups. In contrast, GO-Skill maintains comparable performance across both setups, particularly in the Near-Optimal case, where its performance improves with more tasks rather than fewer. This suggests that GO-Skill effectively leverages the richer shared knowledge to develop more robust skills as the task count grows. In addition, MH-GO-Skill, using a multi-head skill model, mitigates gradient conflicts between different skills, further improving performance.

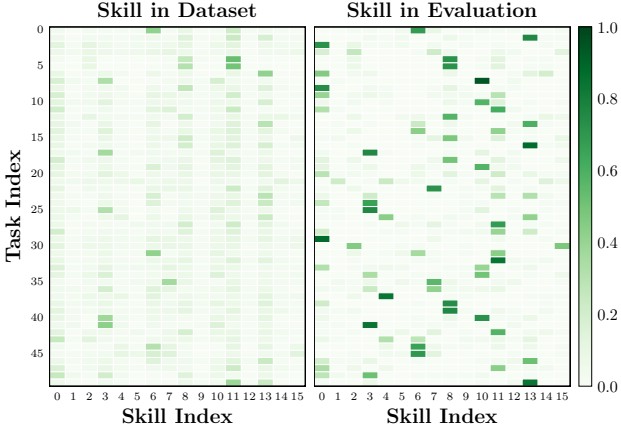

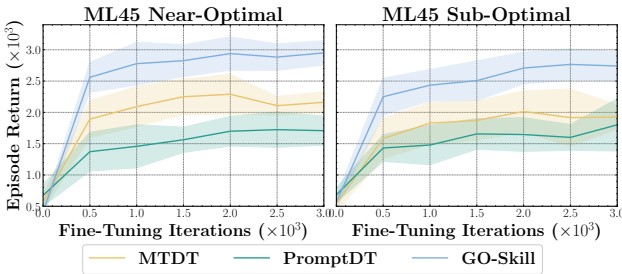

Figure 8. Fine-tuning on five new tasks under Near-Optimal and Sub-Optimal settings in the ML45 setup demonstrates that GO-Skill is highly effective for transfer scenarios.

Figure 7. Visualization of skill distribution in the dataset and evaluation under the Near-Optimal setting in the MT50 setup. This result reveals the imbalance in skill classes, where some skills are widely used across tasks, while others are unique to a small subset. Notably, GO-Skill demonstrates the ability to learn optimal skill combinations from data containing sub-optimal trajectories.

selecting four critical hyper-parameters. Figure 6 illustrates the ablation results under the Near-Optimal setting in the MT50 setup. Across a wide range of hyper-parameter selections, our approach consistently outperforms baseline methods. Among these, the skill horizon exerts the most significant impact on performance. Short horizons fail to provide adequate temporal abstraction, while excessively long horizons increase the informational demands on the skill space, leading to slower convergence. The extraction training iterations also play a pivotal role. Too few iterations are insufficient to extract a valid set of skills, whereas too many iterations extend the total training time. Based on our findings, we recommend halting skill extraction once the skill divisions have largely converged and then proceeding to skill enhancement and skill-based policy learning. Our insights suggest the following recommended settings: skill horizon $H = 10$, codebook size $M = 16$, code embedding size $|\mathcal{Z}| = 64$, and extraction iteration $t_e = 3e4$ (30% of total iteration). These hyper-parameter settings collectively contribute to the superior performance.

### 5.4. Skill Abstraction Visualization

As shown in Figure 7, the left panel visualizes the distribution of skills corresponding to various tasks in the dataset. In contrast, the right panel visualizes the distribution employed by the agent across different tasks during evaluation. Notably, the skill class imbalance exists within the dataset and during evaluation. In addition, each task in the dataset contains fragments of trajectories for multiple skills, but specific skills are not employed during evaluation. Despite

this, the data associated with these unused skills can support other tasks in leveraging those skills effectively. For instance, while the dataset for Task 2 (bin-picking-v2) contains limited data for Skill 0, the agent successfully utilizes Skill 0 to complete the task during evaluation. Further visualization of the learned skill set is provided in Appendix C.

### 5.5. Fine-Tuning on New Tasks

The MTRL model is typically employed as a base model that requires minimal fine-tuning to adapt to new tasks. We utilize the ML45 benchmark in MetaWorld for evaluation. We first pre-train the model for 100,000 iterations on the original 45 tasks and then fine-tune it for 3,000 on 5 new tasks. *Near-Optimal* and *Sub-Optimal* represent the quality of the pre-training dataset, while the fine-tuning data consists exclusively of Near-Optimal expert trajectories. As shown in Figure 8, the results demonstrate that regardless of the quality of the pre-training dataset, GO-Skill effectively extracts a valid set of skills that can be efficiently adapted to the new task set with minimal fine-tuning. Moreover, we find that MTDT outperforms PromptDT in fine-tuning scenarios, as PromptDT generally requires more training iterations to converge on new tasks.

## 6. Conclusion and Discussion

In this work, we propose Goal-Oriented Skill Abstraction (GO-Skill), a novel approach inspired by the efficient knowledge abstraction observed in human learning. Leveraging the goal-oriented skill representation, we learn a skill model to extract reusable skills and develop a skill-based policy to utilize these skills effectively. To address the challenge of skill class imbalance, we introduce a skill enhancement phase and incorporate focal loss into policy learning. Rigorous empirical evaluations on the offline MTRL benchmark demonstrate that GO-Skill achieves superior performance.

**Limitations and Future Works.** We present an innovative approach in offline MTRL, achieving state-of-the-art per-

formance across various tasks. However, its effectiveness relies on the predefined skill horizon and the size of the skill set. Future research could explore methods for dynamically extending the skill set and adaptively determining the skill horizon to enhance flexibility and scalability. In addition, investigating more robust goal-oriented representation emerges as another important direction for future endeavors.

## Acknowledgements

This work is supported in part by the Strategic Priority Research Program of the Chinese Academy of Sciences (Grant No. XDA0480200), the National Science and Technology Major Project (2022ZD0116401), the Natural Science Foundation of China (Grant Nos. 62222606 and 61902402), and the Key Research and Development Program of Jiangsu Province (Grant No. BE2023016).

## Impact Statement

This paper presents work that aims to advance the field of Offline Multi-Task Reinforcement Learning. There are many potential societal consequences of our work, none of which we feel must be specifically highlighted here.

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

# A. Environment Details

The MetaWorld benchmark (Yu et al., 2020b) consists of a diverse set of 50 distinct manipulation tasks, all unified by shared dynamics. These tasks involve a Sawyer robot manipulating a variety of objects, each with unique shapes, joints, and connectivity properties. In its original configuration, the source tasks are defined with fixed goals, which restrict the policy's ability to generalize across tasks of the same type but with varying goal conditions. To address this, we extend these tasks to a random-goal setting, similar to the approach taken in prior work (Yang et al., 2020; He et al., 2023; Hu et al., 2024), where both objects and goals are reset randomly at the start of each episode. We experiment with three different setups: (1) **MT50**, which includes the full set of 50 tasks, (2) **MT30**, a subset containing 30 operational tasks from MT50, and (3) **ML45**, where 45 tasks are used for pre-training and the remaining 5 tasks are used for fine-tuning evaluation. A detailed breakdown of the skill sets is provided in Table 3. As MetaWorld is an evolving benchmark, our experiments are conducted on the stable version, MetaWorld-V2 [1].

For the offline dataset, we follow the approach outlined by He et al. (2023), using the Soft Actor-Critic (SAC) algorithm (Haarnoja et al., 2018) to train a separate policy for each task until convergence. This generates a dataset consisting of 2 million transitions per task, drawn from the SAC replay buffer. These transitions represent the entire set of samples observed during training until the policy's performance stabilizes. We define two distinct dataset settings: (1) **Near-Optimal**, which includes the complete experience (100M transitions) from random to expert-level performance in SAC-Replay, and (2) **Sub-Optimal**, which contains the first 50% (50M transitions) of the near-optimal dataset for each task, where the proportion of expert-level data is significantly lower.

*Table 3.* Detailed task set in different MetaWorld setups.

| **MetaWorld MT30** | |
|---|---|
| **Training tasks** | [ "basketball-v2", "bin-picking-v2", "button-press-topdown-v2", "button-press-v2", "button-press-wall-v2", "coffee-button-v2", "coffee-pull-v2", "coffee-push-v2", "dial-turn-v2", "disassemble-v2", "door-close-v2", "door-lock-v2", "door-open-v2", "door-unlock-v2", "drawer-close-v2", "drawer-open-v2", "faucet-close-v2", "faucet-open-v2", "hand-insert-v2", "handle-press-side-v2", "handle-press-v2", "handle-pull-side-v2", "handle-pull-v2", "lever-pull-v2", "peg-insert-side-v2", "pick-out-of-hole-v2", "pick-place-wall-v2", "push-back-v2", "push-v2", "reach-v2" ] |

| **MetaWorld MT50** | |
|---|---|
| **Training tasks** | All tasks in MetaWorld ML45 including pre-training and fine-tuning. |

| **MetaWorld ML45** | |
|---|---|
| **Pre-training tasks** | [ "assembly-v2", "basketball-v2", "button-press-topdown-v2", "button-press-topdown-wall-v2", "button-press-v2", "button-press-wall-v2", "coffee-button-v2", "coffee-pull-v2", "coffee-push-v2", "dial-turn-v2", "disassemble-v2", "door-close-v2", "door-open-v2", "drawer-close-v2", "drawer-open-v2", "faucet-close-v2", "faucet-open-v2", "hammer-v2", "handle-press-side-v2", "handle-press-v2", "handle-pull-side-v2", "handle-pull-v2", "lever-pull-v2", "peg-insert-side-v2", "peg-unplug-side-v2", "pick-out-of-hole-v2", "pick-place-v2", "pick-place-wall-v2", "plate-slide-back-side-v2", "plate-slide-back-v2", "plate-slide-side-v2", "plate-slide-v2", "push-back-v2", "push-v2", "push-wall-v2", "reach-v2", "reach-wall-v2", "shelf-place-v2", "soccer-v2", "stick-pull-v2", "stick-push-v2", "sweep-into-v2", "sweep-v2", "window-close-v2", "window-open-v2" ] |
| **Fine-tuning tasks** | [ "bin-picking-v2", "box-close-v2", "door-lock-v2", "door-unlock-v2", "hand-insert-v2" ] |

# B. Implementation Details

We implement all experiments using the Prompt-DT (Xu et al., 2022) codebase[2], and access the MetaWorld environment to this framework.

### B.1. Details on Computational Resources

We use NVIDIA Geforce RTX 3090 GPU for training and AMD EPYC 7742 64-Core Processor for evaluation with the environments. The training duration for each model is typically takes around 9 hours in the MT50 setup and approximately 8 hours in the MT30 setup. Since each environment is trained five times with different seeds, the total training time is generally multiplied by five.

---

[1] https://github.com/Farama-Foundation/Metaworld/tree/v2.0.0
[2] https://github.com/mxu34/prompt-dt

## B.2. Details on Hyper-Parameters

We present the common hyper-parameters in Table 4 and the additional hyper-parameters for GO-Skill in Table 5. Notably, the total number of iterations for the baselines is 1e5. GO-Skill first performs skill extraction using 3e4 iterations, followed by parallel iterations of skill enhancement and policy learning for 7e4.

*Table 4.* Common hyper-parameters configuration of GO-Skill and baselines.

| Hyper-parameter | Value |
| --- | --- |
| Number of layers | 6 |
| Number of attention heads | 8 |
| Embedding dimension | 256 |
| Nonlinearity function | ReLU |
| Batch size | 8 per task |
| Context length | 20 |
| Prompt Length | 10 |
| Dropout | 0.1 |
| learning rate | 3e-4 |
| Total iterations | 1e5 |

*Table 5.* Additional hyper-parameters configuration of GO-Skill.

| Hyper-parameter | Value |
| --- | --- |
| Skill horizon | 10 |
| Skill set (codebook) size | 16 |
| Goal/skill embedding dimension | 64 |
| Skill extraction iterations | 3e4 |
| Skill enhancement iterations | 7e4 |
| Skill-based policy learning iterations | 7e4 |
| Focusing parameter | 2 |

## C. Additional Visualization

As shown in Figure 9, we visualize the set of skills learned by GO-Skill under the Near-Optimal setting in the MT50 setup. Across various tasks, GO-Skill effectively learns a shared skill abstraction. For instance, in tasks requiring the grasping of items with unique shapes, joints, and connectivity properties, GO-Skill applies the same grabbing skill across all variations.

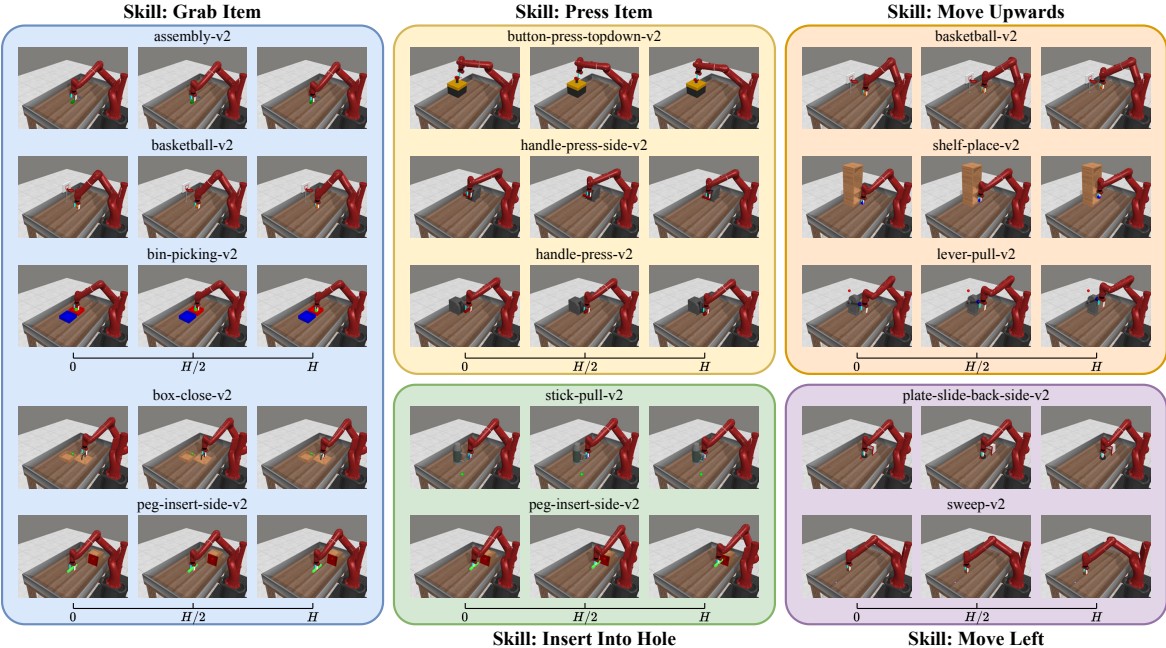

*Figure 9.* Visualization of the five skills in GO-Skill. The three images represent the states within the $H$-horizon.

## D. Single-Task Performance

We present the detailed performance evaluation of GO-Skill on the MT50 setup in Section 5.2, with results for each task provided in Table 6.

*Table 6.* Evaluated mean episode return of GO-Skill for each task in the MT50 setup. The model is trained for 1e5 iterations, with each run evaluated over 50 episodes. Reported values represent the mean and standard deviation across 5 different seeds.

| Task Name | Near-Optimal | Sub-Optimal |
|---|---|---|
| assembly-v2 | $2366.7_{\pm 1082.3}$ | $630.8_{\pm 20.1}$ |
| basketball-v2 | $3634.5_{\pm 658.1}$ | $2774.4_{\pm 326.8}$ |
| bin-picking-v2 | $4137.9_{\pm 212.8}$ | $324.7_{\pm 205.6}$ |
| box-close-v2 | $3387.7_{\pm 229.1}$ | $2177.1_{\pm 436.9}$ |
| button-press-topdown-v2 | $2787.5_{\pm 200.4}$ | $1733.5_{\pm 254.5}$ |
| button-press-topdown-wall-v2 | $2773.3_{\pm 206.3}$ | $1728.5_{\pm 267.1}$ |
| button-press-v2 | $2678.5_{\pm 130.8}$ | $2112.9_{\pm 344.6}$ |
| button-press-wall-v2 | $3358.3_{\pm 239.7}$ | $1962.0_{\pm 591.9}$ |
| coffee-button-v2 | $2939.4_{\pm 357.2}$ | $795.0_{\pm 209.2}$ |
| coffee-pull-v2 | $1421.6_{\pm 519.5}$ | $113.4_{\pm 36.6}$ |
| coffee-push-v2 | $569.1_{\pm 119.2}$ | $220.6_{\pm 156.4}$ |
| dial-turn-v2 | $2872.3_{\pm 202.7}$ | $1814.8_{\pm 297.7}$ |
| disassemble-v2 | $923.2_{\pm 445.4}$ | $376.3_{\pm 121.0}$ |
| door-close-v2 | $4322.7_{\pm 17.7}$ | $4218.9_{\pm 215.8}$ |
| door-lock-v2 | $3428.4_{\pm 194.1}$ | $2819.3_{\pm 331.4}$ |
| door-open-v2 | $3473.2_{\pm 180.4}$ | $2177.4_{\pm 214.1}$ |
| door-unlock-v2 | $3722.3_{\pm 148.3}$ | $3028.0_{\pm 398.0}$ |
| drawer-close-v2 | $4717.6_{\pm 64.2}$ | $4751.3_{\pm 60.5}$ |
| drawer-open-v2 | $2918.8_{\pm 406.9}$ | $2028.3_{\pm 205.7}$ |
| faucet-close-v2 | $4481.4_{\pm 124.7}$ | $4168.0_{\pm 253.8}$ |
| faucet-open-v2 | $4527.9_{\pm 60.4}$ | $4570.8_{\pm 14.7}$ |
| hammer-v2 | $3188.7_{\pm 456.1}$ | $2610.9_{\pm 91.5}$ |
| hand-insert-v2 | $2515.9_{\pm 192.0}$ | $857.8_{\pm 281.2}$ |
| handle-press-side-v2 | $4630.1_{\pm 66.9}$ | $4103.8_{\pm 229.3}$ |
| handle-press-v2 | $4281.8_{\pm 149.9}$ | $3533.4_{\pm 237.5}$ |
| handle-pull-side-v2 | $3715.4_{\pm 447.5}$ | $1751.2_{\pm 733.0}$ |
| handle-pull-v2 | $3451.3_{\pm 489.3}$ | $2549.2_{\pm 493.9}$ |
| lever-pull-v2 | $2819.0_{\pm 447.5}$ | $1424.6_{\pm 220.8}$ |
| peg-insert-side-v2 | $4096.6_{\pm 217.9}$ | $2391.7_{\pm 642.8}$ |
| peg-unplug-side-v2 | $3197.6_{\pm 451.0}$ | $170.4_{\pm 43.8}$ |
| pick-out-of-hole-v2 | $3044.0_{\pm 162.1}$ | $2047.8_{\pm 147.9}$ |
| pick-place-v2 | $358.6_{\pm 409.1}$ | $19.3_{\pm 14.6}$ |
| pick-place-wall-v2 | $957.6_{\pm 767.7}$ | $545.4_{\pm 506.2}$ |
| plate-slide-back-side-v2 | $4333.4_{\pm 210.4}$ | $3210.5_{\pm 558.2}$ |
| plate-slide-back-v2 | $4374.5_{\pm 305.1}$ | $3518.5_{\pm 669.1}$ |
| plate-slide-side-v2 | $2925.9_{\pm 179.4}$ | $1816.4_{\pm 328.4}$ |
| plate-slide-v2 | $4414.0_{\pm 97.0}$ | $3393.5_{\pm 340.5}$ |
| push-back-v2 | $2186.2_{\pm 258.8}$ | $409.2_{\pm 208.5}$ |
| push-v2 | $1114.9_{\pm 80.5}$ | $546.2_{\pm 265.8}$ |
| push-wall-v2 | $4093.3_{\pm 193.0}$ | $2280.6_{\pm 223.4}$ |
| reach-v2 | $3510.0_{\pm 259.7}$ | $2404.2_{\pm 688.6}$ |
| reach-wall-v2 | $4103.3_{\pm 202.3}$ | $3714.4_{\pm 164.3}$ |
| shelf-place-v2 | $3532.0_{\pm 303.0}$ | $1365.8_{\pm 367.2}$ |
| soccer-v2 | $1013.8_{\pm 219.5}$ | $967.6_{\pm 367.6}$ |
| stick-pull-v2 | $3408.4_{\pm 234.5}$ | $3591.1_{\pm 309.9}$ |
| stick-push-v2 | $3785.7_{\pm 408.5}$ | $2265.6_{\pm 464.5}$ |
| sweep-into-v2 | $4030.7_{\pm 268.7}$ | $3031.6_{\pm 546.5}$ |
| sweep-v2 | $4100.4_{\pm 227.7}$ | $3760.8_{\pm 208.1}$ |
| window-close-v2 | $2774.6_{\pm 496.1}$ | $2174.8_{\pm 367.2}$ |
| window-open-v2 | $2557.1_{\pm 242.0}$ | $1206.6_{\pm 180.7}$ |

