# OpenReview forum: "Goal-Oriented Skill Abstraction for Offline Multi-Task Reinforcement Learning"
_ICML.cc/2025/Conference — ICML 2025 poster_

### Official Review · Reviewer_FkAC · 2025-03-13

**Overall Recommendation:** 3

**Summary:**

The paper proposes Goal-Oriented Skill Abstraction (GO-Skill), a novel method for offline multi-task reinforcement learning (MTRL) that learns a unified policy to solve multiple tasks using pre-collected, task-mixed datasets. It introduces a skill extraction process using a goal encoder, vector quantization (VQ), and a skill decoder to build a discrete skill library from offline trajectories, followed by a skill enhancement phase to address class imbalance. A high-level skill-based policy, implemented as a Prompt Decision Transformer, dynamically combines these skills to tackle specific tasks. Main findings include superior performance on the MetaWorld benchmark (MT30 and MT50 setups), with average episode returns of 3159.1 (Near-Optimal) and 2123.8 (Sub-Optimal) on MT50, outperforming baselines like MTDT and PromptDT.

**Claims And Evidence:**

The claim that GO-Skill enhances knowledge transfer and task performance via reusable skills is well supported by experiments on MetaWorld MT30 and MT50, showing higher average episode returns.

**Essential References Not Discussed:**

To my knowledge, no essential references are missing. See the section "Relation To Broader Scientific Literature".

**Experimental Designs Or Analyses:**

The experiment compare serveral Decision Transformer-style RL models in MTRL and demonstrates its effectiveness with statistical significant improvement. The qualitative result of ablation study is well-designed and clear.

**Methods And Evaluation Criteria:**

The GO-Skill method, involving goal-oriented skill extraction and hierarchical policy learning, is appropriate for offline MTRL, leveraging offline data to abstract reusable skills.

**Other Comments Or Suggestions:**

See above.

**Other Strengths And Weaknesses:**

Strengths:
1. the paper is well written and clear.
2. Originality in goal-oriented skill abstraction, strong empirical results on MetaWorld, and clear visualizations (e.g., Figure 7). The skill enhancement phase addressing imbalance is a practical innovation.

Weaknesses:
1. Skill Extraction need addtional training cost to mitigate the skill imbalance problems.
2. Over-reliance on MetaWorld limits domain diversity, such as LIBERO.

**Questions For Authors:**

1. What does “dynamic transfer” mean in Section 3.1.

**Relation To Broader Scientific Literature:**

The paper omits discussion of "LIBERO: Benchmarking Knowledge Transfer for Lifelong Robot Learning" by Liu et al. (NeurIPS 2023), which is highly relevant to its contributions in knowledge-sharing for multi-task RL. LIBERO provides a benchmark for evaluating knowledge transfer—both declarative and procedural—in lifelong RL settings.

Given the paper’s focus on sharing knowledge across RL tasks, referencing LIBERO could contextualize its approach against a standardized benchmark. This omission counld limits the ability to assess how the proposed method compares to or builds upon established knowledge transfer frameworks in RL.

**Theoretical Claims:**

The paper does not present formal theoretical claims or proofs. It includes serveral equations defining the loss objective to optimize the proposed model’s performance and basic formulation of model framework. As such, there are no proofs to verify.

---

> ### Author Rebuttal · Authors · 2025-03-31
>
> Thank you very much for your constructive feedback and acknowledgment of our efforts. Following are our responses to all your concerns.
>
> > Skill Extraction need addtional training cost to mitigate the skill imbalance problems.
>
> As detailed in Appendix B.2, we ensure that the total number of training iterations remains consistent across all methods. Specifically, the sum of stage 1 iterations (Skill Extraction) and stage 2 iterations (Skill Enhancement and Skill-Based Policy Learning) in GO-Skill is equal to the total iterations of the baselines. Importantly, since Skill Enhancement and Skill-Based Policy Learning are entirely independent learning processes, they are trained in parallel during the second stage of GO-Skill. As a result, no extra training time or iterations are introduced compared to the baselines. This design allows us to incorporate skill balancing mechanisms without increasing overall computational cost or compromising fairness in comparison.
>
> > Over-reliance on MetaWorld limits domain diversity, such as LIBERO.
>
> Thank you for suggesting the LIBERO benchmark. However, we would like to clarify that LIBERO is fundamentally a multi-task imitation learning benchmark rather than a MTRL benchmark. The dataset only provides a limited number of expert demonstrations (50 trajectories per task) and lacks reward information (only a binary success signal). This setup does not fully capture one of the key advantages of GO-Skill: suboptimal trajectories from certain tasks can still facilitate skill learning and potentially benefit other tasks leveraging the same skill. We have analyzed this phenomenon in Section 5.4 of the main paper.
>
> Nevertheless, we adapted our approach to the LIBERO setting by employing an imitation learning paradigm, removing the return-conditioning in DT-based models, and comparing GO-Skill, PromptDT, and MTDT under this framework. Additionally, since LIBERO is a vision-based environment, we utilized R3m\[1] to encode image observations into vector representations. We use the LIBERO90 setting, and the experimental results are as follows:
>
> | Method   | LIBERO           |
> | -------- | ---------------- |
> | MTDT     | \$9.9 \pm 3.5\$  |
> | PromptDT | \$17.6 \pm 3.7\$ |
> | GO-Skill | \$29.4 \pm 4.0\$ |
>
> From the experimental results, it can be seen that the explicit knowledge sharing provided by skill abstraction can also help agent to learn multiple tasks quickly in a multi-task imitation learning scenario.
>
> Furthermore, based on suggestions from other reviewers, we also conducted few-shot experiments on *Cheetah-Vel* and *Ant-Dir*, following the PromptDT setup. The results are shown below:
>
> | Method   | Cheetah-Vel        | Ant-Dir            |
> | -------- | ------------------ | ------------------ |
> | MTDT     | \$-158.6 \pm 9.8\$ | \$168.3 \pm 20.7\$ |
> | PromptDT | \$-39.5 \pm 4.2 \$ | \$398.9 \pm 39.1\$ |
> | GO-Skill | \$-40.8 \pm 3.6 \$ | \$411.2 \pm 32.5\$ |
>
> As shown in the table, GO-Skill achieves comparable performance to PromptDT. This is expected because these environments are relatively simple, and the high similarity between different tasks causes both GO-Skill and PromptDT to reach the performance ceiling of these benchmarks.
>
> \[1] Nair, Suraj, et al. R3m: a universal visual representation for robot manipulation. CoRL 2022.
>
> > What does "dynamic transfer" mean in Section 3.1.
>
> "Dynamic transfer" refers to the MDP’s transition dynamics, i.e., how the state evolves. The key distinction here is that this dynamic transition describes the **state change after H steps of decision-making**, rather than a single-step transition. If this phrasing has caused any misunderstanding, we will revise it to "H-step dynamic transfer" in the revised manuscript to clarify the meaning.
>
> **Thank you once again for your valuable feedback. If you need further elaboration or additional points to include in the response, we welcome further discussion to ensure everything is clear and satisfactory.**

---

### Official Review · Reviewer_FMhW · 2025-03-13

**Overall Recommendation:** 4

**Summary:**

This paper introduces GO-Skill, a novel hierarchical framework for offline multi-task reinforcement learning. The approach decouples learning into two components:
- A low-level action predictor ("skill-decoder transformer") that generates actions based on a given skill prompt
- A high-level skill predictor ("skill-based decision transformer") built upon the PromptDT architecture that selects appropriate skills to serve as prompts for the skill-decoder

The main innovation lies in the skill representation: embeddings that capture state variations over a fixed horizon $H$. The authors evaluate their approach on the MetaWorld benchmark, demonstrating consistent improvements over relevant baseline methods.

**Claims And Evidence:**

The claims made in the submission are supported by clear and convincing evidence. The authors demonstrate GO-Skill's greater performance across multiple MetaWorld benchmarks (MT30, MT50) under different data conditions through relevant comparisons with previous methods. The ablation studies validate the contribution of each component (goal-oriented representation, vector quantization, skill enhancement). The paper provides interesting qualitative analysis, with visualizations of skill distributions proving method's ability to learn transferable skills, and examples of skills depicted in Appendix. The experimental methodology is sound: the paper uses multiple random seeds for statistical significance and provides detailed per-task results in the supplementary material.

**Essential References Not Discussed:**

To the best of my knowledge, all the releva related works are cited and discussed.

**Experimental Designs Or Analyses:**

The experimental design is solid and well-organized. The authors test GO-Skill by comparing it with six different baseline approaches across four test conditions (MT30/MT50 combined with Near-Optimal/Sub-Optimal data). They also include several ablation studies that examine the importance of different parts of their system, such as how the skill model is designed, how they handle the problem of skill imbalance, and how different parameter settings affect performance. To help understand how their method works, they include helpful visualizations showing how skills are distributed and how well their system adapts to new tasks during fine-tuning. I also found the skill visualizations from Appendix C particularly interesting.

**Methods And Evaluation Criteria:**

The proposed methods and evaluation criteria are appropriate for the problem. The MetaWorld benchmark provides a diverse set of robotic manipulation tasks suitable for evaluating multi-task reinforcement learning. The authors evaluate their approach under different conditions (including varying numbers of tasks and dataset quality) and compare against relevant state-of-the-art approaches. The ablation studies effectively isolate the contributions of different components of the GO-Skill architecture.

**Other Comments Or Suggestions:**

Minor issues:

- Line 132, 2nd column: Technically, the goal encoder should be a function from $\mathcal{S}$ to $\mathcal{Z}$ ($\mathcal{S} \rightarrow \mathcal{Z}$).
- In Algorithm 2: The "get prompt" line appears outside the "for each task" loop, which seems counterintuitive since prompts would typically be task-dependent.

**Other Strengths And Weaknesses:**

Strengths:

- The paper is well-structured and clearly written, making complex technical concepts accessible.
- The approach has a strong conceptual motivation inspired by human learning.
- The skill visualization provides intuitive understanding of what the model is learning.
- The method appears to effectively leverage sub-optimal data by extracting useful skill fragments.

Weaknesses:

- The approach has only been evaluated on a single benchmark (MetaWorld), which limits understanding of its generalizability to other domains.
- The trajectory-level state difference representation of skills may not capture all relevant aspects of useful skills.
- The fixed skill horizon is a limitation that the authors acknowledge, as different skills naturally operate at different timescales.
- The evaluation doesn't include locomotion tasks like "Cheetah-vel" and "Ant-dir" that were studied in related work like PromptDT.

**Questions For Authors:**

1. In Algorithm 2, why is the "get prompt" line not included in the "for each task" loop? Since prompts are typically task-specific, this seems counterintuitive. Could you clarify how prompts are handled across different tasks?
2. Have you considered how your approach could be extended to learn skills with different temporal horizons? Some manipulations (like grabbing) might occur over shorter horizons than others (like transporting).
3. How might your approach perform on locomotion tasks like those studied in the PromptDT paper (e.g., "Cheetah-vel" and "Ant-dir")? These environments have different dynamics from manipulation tasks, which could test the generality of your skill abstraction method.

**Relation To Broader Scientific Literature:**

The paper differentiates itself from previous methods that mainly work on low-level action imitation. Instead, this work puts emphasis on higher-level skill abstraction, which takes inspiration from how humans learn. This approach allows the system to identify reusable skills across different tasks.

**Theoretical Claims:**

Not Applicable (there are no theoretical claims).

---

> ### Author Rebuttal · Authors · 2025-03-31
>
> Thank you very much for your constructive feedback and acknowledgment of our efforts. Following are our responses to all your concerns.
>
> > Perform on locomotion tasks "Cheetah-vel" and "Ant-dir"
>
> We followed the experimental setup of PromptDT and conducted few-shot experiments on the *Cheetah-Vel* and *Ant-Dir* environments. The detailed results are as follows:
>
> | Method   | Cheetah-Vel        | Ant-Dir            |
> | -------- | ------------------ | ------------------ |
> | MTDT     | \$-158.6 \pm 9.8\$ | \$168.3 \pm 20.7\$ |
> | PromptDT | \$-39.5 \pm 4.2 \$ | \$398.9 \pm 39.1\$ |
> | GO-Skill | \$-40.8 \pm 3.6 \$ | \$411.2 \pm 32.5\$ |
>
> As shown in the table, GO-Skill achieves comparable performance to PromptDT. This is expected because these environments are relatively simple, and the high similarity between different tasks causes both GO-Skill and PromptDT to reach the performance ceiling of these benchmarks.
>
> Furthermore, based on suggestions from other reviewers, we also conducted experiments on another multi-task imitation learning benchmark *LIBERO*. Due to space constraints, we kindly refer you to our response to **Reviewer FkAC** for detailed information. For convenience, we recommend using **Ctrl+F** to search for "**LIBERO benchmark**" to quickly locate the relevant response.
>
> > The trajectory-level state difference representation of skills may not capture all relevant aspects of useful skills.
>
> We appreciate your thoughtful observation. Indeed, no single representation can capture all aspects of useful skills. GO-Skill adopts a goal-based representation to define skills in a task-agnostic manner. As shown in our ablation study in Section 5.3, this representation outperforms action-based alternatives. We believe this is because multiple action sequences can lead to the same state transition, and abstracting skills based on goals allows the agent to generalize across these variations, leading to more transferable and reusable skills.
>
> We agree that goal-based representation may not always be optimal, but our framework is flexible and can accommodate alternative skill representations. We thank you for highlighting this direction, and we plan to further explore more expressive representations as part of our future work.
>
> > How GO-Skill could be extended to learn skills with different temporal horizons?
>
> Thank you for raising this important point. While the current GO-Skill implementation adopts a fixed temporal horizon for simplicity and consistency, the framework itself is **fully compatible with variable-length skills**.
>
> To support variable-length skills, the primary change would be in the **skill decoder**, which can be extended to learn a terminate signal to determine when to exit a skill. The key challenge lies in **how to extract and define variable-length skills** during the offline skill discovery phase. As mentioned in our Future Work Section, this is a central focus of our ongoing research. We are currently exploring approach by applying thresholds on state changes to identify skill. When the state change exceeds a certain threshold, it indicates the need to switch to a new skill.
>
> We believe enabling dynamic skill lengths will further enhance the expressiveness and adaptability of GO-Skill, and we look forward to sharing these results in future work.
>
> > Add a reference to Appendix C in results section.
>
> Thank you for your suggestion and for recognizing the value of the visualization results. We will add a reference to Appendix C in the results section in the revised manuscript.
>
> > "get prompt" appears outside the "for each task" loop in Alg.2.
>
> Thank you for pointing out this mistake. This is indeed a writing error, and we will correct it in the revised manuscript.
>
> Additionally, to clarify the prompt selection process: similar to PromptDT, we construct a Prompt Set for each task with the top-N highest return trajectories (N=4 in our experiments). Each time "get prompt" is called, a trajectory is randomly sampled from the corresponding task’s Prompt Set.
>
> > Goal encoder should be $\mathcal{G}: \mathcal{S} \rightarrow \mathcal{Z}$
>
> Thank you very much for pointing out the error. We will correct it in the revised manuscript.
>
> **Thank you once again for your valuable feedback. If you need further elaboration or additional points to include in the response, we welcome further discussion to ensure everything is clear and satisfactory.**

---

> > ### Comment · Reviewer_FMhW · 2025-04-03
> >
> > Thank you very much for your responses and the additional results & analysis. They definitely address all the interrogations I had. Furthermore, I completely agree with the suggested changes.

---

> > > ### Author Response · Authors · 2025-04-03
> > >
> > > Dear Reviewer,
> > >
> > > Thank you for your thoughtful feedback and encouraging response. We are glad our response helped address your concerns. We will incorporate the suggested changes into the final version and believe they also help to resolve related points raised by other reviewers. We sincerely appreciate all the constructive comments and the opportunity to further improve the paper.
> > >
> > > Best regards,
> > >
> > > All authors

---

### Official Review · Reviewer_BwoC · 2025-03-20

**Overall Recommendation:** 4

**Summary:**

This paper presents Goal-Oriented Skill Abstraction (GO-Skill), an approach aimed at enhancing knowledge transfer in offline multi-task reinforcement learning (MTRL). GO-Skill extracts reusable skills from task-mixed offline datasets through goal-oriented representations combined with vector quantization, creating a discrete skill library. To handle imbalances among skill categories, the method includes a refinement phase. These skills are then utilized via a hierarchical policy, demonstrating performance improvements on MetaWorld benchmark tasks compared to several baselines.

**Claims And Evidence:**

All the claims and architectural choices are well-motivated and validated by ablations.

**Essential References Not Discussed:**

The authors overlooked important related offline MTRL baselines, particularly the conservative data-sharing approach proposed by Yu et al., which is essential for the proper contextualization of their results.

**Experimental Designs Or Analyses:**

Overall, the experimental setup is thorough, clearly comparing GO-Skill to several relevant baseline methods across multiple scenarios (MT30, MT50, near-optimal and sub-optimal datasets).

However, an important baseline from existing literature, such as data-sharing methods described in "Conservative Data Sharing for Multi-Task Offline Reinforcement Learning" by Yu et al., is notably missing.

Furthermore, the paper contains only one category of MTRL benchmark, Metaworld, which is not sufficient to justify all the components introduced in this paper.

**Methods And Evaluation Criteria:**

The methods and evaluation metrics used are appropriate for offline multi-task reinforcement learning problems. Using the widely accepted MetaWorld benchmark provides valid grounds for comparison with baselines. However, results are reported using average return rather than success rate, which is the standard metric in this domain. I'd recommend including the success rate as the primary comparison metric.

**Other Comments Or Suggestions:**

More environments should be introduced, at least for the baseline comparison.

**Other Strengths And Weaknesses:**

## Strengths
- Intuitive and well-motivated skill abstraction inspired by human learning.
- Strong empirical evidence of improved performance against baselines and ablations.

## Weaknesses:
- The method is quite complex and it is not clear whether all the components are necessary in general. I'd suspect for different MTRL task suites, not all the proposed contributions would be necessary.
- Lacks success rate, which limits the comparability of the results.
- Only a single environment setup is used — Metaword.

**Questions For Authors:**

1. How is the hierarchical skill abstraction fundamentally different from previously proposed methods in offline RL and multi-task learning?

**Relation To Broader Scientific Literature:**

The authors effectively place their work within the existing offline RL and MTRL literature, building on prior approaches such as Decision Transformers, Prompt-DT, and vector quantization, clearly situating their contribution.

**Theoretical Claims:**

The paper does not include explicit theoretical claims or proofs.

---

> ### Author Rebuttal · Authors · 2025-03-31
>
> Thank you very much for your constructive feedback and acknowledgment of our efforts. Following are our responses to all your concerns.
>
> >Overlook offline MTRL baseline CDS.
>
> Thank you for pointing out the omission of the CDS, which is indeed a significant contribution to offline MTRL. We will cite and discuss CDS more thoroughly in the revised manuscript.
>
> CDS focuses on a conservative data-sharing strategy, selectively utilizing data from other tasks to benefit learning on a given target task. In contrast, GO-Skill approaches **knowledge sharing at a higher level of abstraction** by discovering reusable skills that can be naturally composed across tasks. Moreover, these two methods could be integrated, where the high-level skill-based policy in GO-Skill can be enhanced by applying conservative filtering to determine which skill-based transitions to share across tasks as proposed in CDS, leading to more robust policy learning. We believe this hybrid direction holds promise and represents an exciting avenue for future work.
>
> Unfortunately, CDS’s codebase and experimental data are not publicly available, making a direct comparison difficult. Moreover, reimplementing such a method without official reference results for validation could risk an unfair or inaccurate evaluation. Nonetheless, we are currently emailing the authors to request the necessary resources and guidance for reproduction. If successful, we will experimentally explore the potential complementarity between CDS and GO-Skill, and we are excited to investigate the results in the final version.
>
> >Only a single environment setup is used — Metaword.
>
> Based on suggestions from other reviewers, we have conducted experiments on two additional benchmarks beyond MetaWorld. One is the locomotion task suite (*Cheetah-Vel* and *Ant-Dir*) used in PromptDT, and the other is *LIBERO*, a multi-task imitation learning benchmark.
>
> Due to space constraints, we kindly refer you to our response to **Reviewer FkAC** for detailed information. For convenience, we recommend using **Ctrl+F** to search for "**LIBERO benchmark**" to quickly locate the relevant response.
>
> >The method is quite complex and it is not clear whether all the components are necessary in general. I'd suspect for different MTRL task suites, not all the proposed contributions would be necessary.
>
> As noted above, we also evaluate GO-Skill on additional benchmarks. The results demonstrate the general applicability of GO-Skill, supporting that it is not tailored to a specific environment.
>
> In addition, our method is not conceptually complex. It consists of two main components: the skill model, which is shared across all tasks and generates actions to interact with the environment, and the skill-based policy, which completes specific tasks by composing different skills. In our ablation study, we systematically analyze both the skill model architecture and the methods for handling skill imbalance, demonstrating that each component contributes to overall performance.
>
> > Lacks success rate, which limits the comparability of the results.
>
> We chose to report return as the primary evaluation measure in MetaWorld because it provides a more informative metric of task performance. For example, in the 'drawer-close' task, the environment considers the task successful once the drawer is closed past a certain threshold — even if it is not fully closed. However, the agent can continue to receive additional rewards for closing the drawer further, indicating a higher degree of task completion. In such cases, success rate acts as a binary threshold, while return reflects the quality and completeness of the agent’s behavior.
>
> In addition, we conducted additional experiments and report the success rate of GO-Skill as follows:
>
> |MT30 Near-Optimal|MT30 Sub-Optimal|MT50 Near-Optimal|MT50 Sub-Optimal|
> |-|-|-|-|
> |$85.5\pm1.0$|$62.8\pm1.4$|$82.8\pm1.0$|$58.2\pm1.4$|
>
> > How is the hierarchical skill abstraction fundamentally different from previously proposed methods?
>
> Our **skill-based MTRL framework** offers a fundamentally different perspective from prior methods. Specifically, most prior multi-task RL methods achieve generalization through **implicit knowledge sharing** through various parameter-sharing strategies. In contrast, GO-Skill adopts an **explicit knowledge sharing** approach by directly extracting reusable skills from offline data. A key advantage of this explicit skill abstraction is its ability to better leverage suboptimal data. In offline settings, a suboptimal skill in one task may serve as an optimal skill for another, allowing GO-Skill to effectively master useful behaviors across tasks. This leads to more efficient skill acquisition and enhances generalization across diverse multi-task scenarios.
>
> **Thank you once again for your valuable feedback. If you need further elaboration or additional points to include in the response, we welcome further discussion to ensure everything is clear and satisfactory.**

---

### Official Review · Reviewer_dNvS · 2025-03-21

**Overall Recommendation:** 3

**Summary:**

This paper proposes a method for offline multi-task reinforcement learning. The main idea is to an approach based on goal-oriented skill abstraction to better learn to extract and reuse skills to transfer to new tasks. Technically, the method utilizes vector quantization to form a discrete skill library. The idea is reasonable, and the paper shows improved performance.

# update after rebuttal

The rebuttal helps address some of the concerns and I remain broadly positive with the paper.

**Claims And Evidence:**

The claims are supported with experimental evidence.

**Essential References Not Discussed:**

No.

**Experimental Designs Or Analyses:**

The experimental and analyses are reasonable and appear correct.

**Methods And Evaluation Criteria:**

The methods are plausible and and the evaluation criteria follow the standard practice.

**Other Comments Or Suggestions:**

The main comparison results shown in Fig. 5 are presented without actual numbers. The y-axis starting from 1.5 makes it difficult to judge the difference of different settings.

**Other Strengths And Weaknesses:**

Strengths:

The method is plausible and well justified.
The paper shows improved performance on standard benchmark datasets.

Weaknesses:

Technical components of the paper are quite standard, so better justification would be beneficial.

**Questions For Authors:**

None.

**Relation To Broader Scientific Literature:**

The proposed method shows improved performance compared with existing multi-task reinforcement learning methods on standard benchmark datasets.

**Theoretical Claims:**

The paper does not make significant theoretical claims.

---

> ### Author Rebuttal · Authors · 2025-03-31
>
> Thank you very much for your constructive feedback and acknowledgment of our efforts. Following are our responses to all your concerns.
>
> > Technical components of the paper are quite standard, so better justification would be beneficial.
>
> We appreciate your comment and acknowledge that certain technical components of our framework, such as vector quantization and transformer-based policy modeling, are standard techniques. However, these choices are made to **validate the effectiveness of our framework**, rather than serving as core contributions. Alternative implementations, such as diffusion-based policies or different discretization methods, could also be applied without altering the fundamental ideas of our approach.
>
> The key contribution of GO-Skill lies in proposing **a novel skill-based MTRL framework** that enables **explicit knowledge sharing** through skill learning and composition, inspired by how humans generalize through modular skills. Furthermore, we introduce **Goal-Oriented Skill Abstraction**, a new method for defining and extracting reusable skills based on goal transitions rather than action sequences or rewards. This task-agnostic representation enhances skill transferability and enables effective skill discovery even from sub-optimal data.
>
> In Section 5, we justify different components through an ablation study. The results show that while standard techniques like VQ contribute to performance improvements, Goal-Oriented Skill provides the most significant boost, highlighting its critical role. In addition, all ablated variants of GO-Skill outperform the baselines, demonstrating the effectiveness of our skill-based framework.
>
> We hope this clarifies that our contributions are primarily conceptual and architectural, and that our use of standard components supports, rather than limits, the novelty of our approach.
>
> > The main comparison results shown in Fig. 5 are presented without actual numbers. The y-axis starting from 1.5 makes it difficult to judge the difference of different settings.
>
> We appreciate your valuable feedback on Fig.5. In the revised manuscript, we will include a table presenting the exact numerical results corresponding to Fig.5 for clarity. Below is the table summarizing the results:
>
> | Method      | MT30 Near-Optimal | MT30 Sub-Optimal | MT50 Near-Optimal | MT50 Sub-Optimal |
> | ----------- | ----------------- | ---------------- | ----------------- | ---------------- |
> | MTDT        | \$2342 \pm 149\$  | \$1690 \pm 64\$  | \$2290 \pm 121\$  | \$1682 \pm 116\$ |
> | MH-MTDT     | \$2434 \pm 123\$  | \$1799 \pm 69\$  | \$2356 \pm 158\$  | \$1786 \pm 84\$  |
> | PromptDT    | \$2497 \pm 138\$  | \$1780 \pm 86\$  | \$2315 \pm 101\$  | \$1787 \pm 105\$ |
> | MH-PromptDT | \$2587 \pm 105\$  | \$1843 \pm 103\$ | \$2478 \pm 57\$   | \$1841 \pm 35\$  |
> | MTDIFF-P    | \$2458 \pm 109\$  | \$1756 \pm 96\$  | \$2387 \pm 91\$   | \$1781 \pm 89\$  |
> | HarmoDT     | \$2874 \pm 84\$   | \$2136 \pm 68\$ | \$2745 \pm 91\$   | \$1921 \pm 104\$ |
> | GO-Skill    | \$3059 \pm 70\$   | \$2168 \pm 73\$  | \$3159 \pm 71\$   | \$2124 \pm 78\$  |
> | MH-GO-Skill | \$3096 \pm 76\$   | \$2175 \pm 67\$  | \$3181 \pm 68\$   | \$2166 \pm 57\$  |
>
> The reason for setting the y-axis starting from 1.5k is to better highlight the performance differences among different methods while maintaining a consistent y-scale across all four settings for easy cross-comparison. In addition, we would greatly appreciate any suggestions you may have for an improved visualization standard, and we will gladly incorporate the necessary adjustments in the revised manuscript accordingly.
>
> **Thank you once again for your valuable feedback. If you need further elaboration or additional points to include in the response, we welcome further discussion to ensure everything is clear and satisfactory.**

---

### Decision · Program_Chairs · 2025-05-01

**Decision:**

Accept (poster)

**Comment:**

This submission received 2 accepts and 2 weak-accepts, and all reviewers agreed that this paper presents a well-motivated and novel approach to offline multi-task reinforcement learning through Goal-Oriented Skill Abstraction. Strengths highlighted include strong empirical results, clear writing, and effective use of skill-based frameworks for knowledge transfer. While initial concerns were raised about reliance on standard techniques, limited benchmarks, missing baselines (CDS), and evaluation metrics, the authors provided thorough rebuttals, additional experiments (LIBERO, Cheetah-Vel, Ant-Dir), and clarifications that addressed most issues. All agreed that this paper demonstrates clear contributions and has been improved through the review process.